# A Bayesian modelling framework for estimating tick-borne pathogen transmission dynamics at the host-tick interface

Younjung Kim[1¤*], Bruno Faivre[2], Thierry Boulinier[3], Célia Sineau[2], Clémence Galon[4], Sara Moutailler[4], Laure Bournez[5], Raphaëlle Métras[1]

**1** Sorbonne Université, INSERM, Institut Pierre Louis d'Épidémiologie et de Santé Publique (IPLESP), Paris, France, **2** Biogéosciences, UMR CNRS, Université Bourgogne-Europe, Dijon, France, **3** CEFE, UMR, CNRS, University of Montpellier, EPHE, IRD, Montpellier, France, **4** ANSES, INRAE, Ecole Nationale Vétérinaire d'Alfort, UMR BIPAR, Laboratoire de Santé Animale, Maisons-Alfort, France, **5** ANSES, Nancy Laboratory for Rabies and Wildlife, Malzéville, France

¤ Current address: Department of Statistics, University of Oxford, Oxford, United Kingdom,
* younjung.kim@stats.ox.ac.uk

## Abstract

Understanding the transmission dynamics of tick-borne pathogens at the host-tick interface is challenged by the presence of multiple pathways for tick infection, including (i) host-to-tick transmission, (ii) tick-to-tick (cofeeding) transmission, and (iii) pre-existing infection through vertical transmission or prior feeding. Assessing parameters governing these pathways is critical for identifying the main transmission drivers and, consequently, key prevention and control points. Here, we developed a Bayesian modelling framework that estimates key parameters describing the probability of each transmission pathway and assesses associated factors, including bird species, tick life stage and engorgement level, by jointly modelling transmission at the host-tick interface using data collected in field studies that sample hosts and their ticks. First, by fitting the model to simulated host-tick infection data, we demonstrated the framework's ability to recover the parameter values underlying these data. Model performance improved significantly when more information was available on variability in cofeeding probability among individual ticks, highlighting the value of testing all collected ticks and recording their spatial distribution on the host in relation to each other. Second, we fitted the model to field data collected at the bird-tick interface in Northeast France in 2023, focusing on *Borrelia garinii*, *B. valaisiana*, and *Anaplasma phagocytophilum* as case pathogens. For all three pathogens studied, models including cofeeding transmission explained the data significantly better than models that did not. Engorgement level was significantly and positively associated with the probability of bird-to-tick transmission for *A. phagocytophilum*. Finally, the estimated parameters, such as the probability of *A. phagocytophilum* infection in birds and the probability of *Borrelia* or *Anaplasma* infection in ticks before feeding, were comparable to values from an external dataset, not used for model fitting. Our framework

**Data availability statement:** Data and code are available at https://github.com/KimYounjung/HostTickPathogenSystem.

**Funding:** This research is supported by French Agence Nationale de la Recherche (MoZArt project, project number ANR-22-CE35-0003 to RM). The funders had no role in study design, data collection and analysis, decision to publish, or preparation of the manuscript.

**Competing interests:** The authors have declared that no competing interests exist.

provides a valuable foundation for future research to understand tick-borne pathogen transmission dynamics based on epidemiological and ecological field data collected at the host-tick interface.

## Author summary

Multiple transmission pathways exist at the tick-host interface, including direct host-to-tick transmission, indirect cofeeding transmission between ticks, and pre-existing infection acquired vertically or during earlier feeding. However, conventional approaches often assume simpler, more direct transmission dynamics, which hampers efforts to understand the transmission dynamics of tick-borne pathogens. We introduce a statistical framework that disentangles these pathways and estimates key parameters using epidemiological and ecological field data collected in field studies that sample ticks and their hosts. Through simulations and fitting a model to field data from birds and ticks for three case pathogens (*Borrelia garinii*, *B. valaisiana*, and *Anaplasma phagocytophilum*), we show that the model can recover true parameter values. Our analyses further indicate that detailed information on tick-to-tick interactions on the host substantially improves the reliability of parameter estimates for a given sample size, providing practical guidance for future field studies applying this framework.

## Introduction

Tick-borne pathogens are maintained in nature through multiple transmission pathways. These include transmission (i) between ticks and vertebrate hosts (host-to-tick or tick-to-host), (ii) between ticks feeding in proximity on the same host (tick-to-tick, i.e., cofeeding), (iii) from female ticks to their eggs (transovarian), and from feeding in previous life stage (transstadial), with their relative contributions varying across different pathogens [1,2]. Key epidemiological data essential for understanding these transmission dynamics include evidence of infection in both hosts and the ticks feeding directly on them, as this information captures direct and simultaneous tick-host-pathogen interactions in their natural environment.

Determining the infection status in ticks is relatively straightforward, as they are competent vectors that likely remain infected after acquiring a pathogen [3]. Consequently, the majority of research has focused on estimating the prevalence of various tick-borne pathogens in ticks, with notable examples including the TBE virus (TBEV) [4], the *Borrelia burgdorferi* sensu lato (s.l.) complex [5], and *Anaplasma phagocytophilum* [6]. In addition to tick infection data, field studies targeting ticks routinely collect information on ticks including tick life stage (e.g., larva, nymph, or adult) and engorgement level, and when ticks are sampled on vertebrate hosts, information on hosts including host species, age, the number of ticks feeding on individual hosts [7–9]. These ecological data can help understand tick-borne pathogen transmission

dynamics by informing analyses of host-tick interactions within a framework designed to reflect the transmission pathways involved.

However, unlike in ticks, determining the infection status in vertebrate hosts presents several significant challenges at multiple levels. First, pathogen persistence levels can vary widely by a variety of factors, including the type of sample used for molecular testing, the time elapsed since infection, the host's condition, and across different pathogens [10,11]. For instance, bacteria within the *Borrelia burgdorferi* sensu lato (s.l.) complex typically establish localised infections in the skin or other tissues rather than systemic infections, making blood an unreliable indicator of active infection [11]. This necessitates more invasive sampling methods, which require significant ethical consideration and sampling efforts. Furthermore, even with skin samples, infections can be missed depending on *Borrelia* genospecies [11] and the extent of *Borrelia* deposition at the sampling site, as demonstrated in studies where *Borrelia* infection was confirmed in xenodiagnostic ticks, despite skin samples testing negative [12,13]. Infections may also remain latent (i.e., non-infectious), depending on the host's condition [12,14]. Second, while serological testing can indicate past infection and potentially protection against certain tick-borne pathogens, the circulation of closely related pathogens (e.g., different *Borrelia* genospecies or flaviviruses) and limited understanding of cross-reactivity between them complicate the interpretation of serological results [15–17]. These challenges underscore the difficulties of directly determining host infection status and understanding their role in pathogen transmission within host-tick interactions.

In the present study, we propose a Bayesian hierarchical latent modelling framework as a complementary and efficient approach to address these challenges. This framework combines multiple sources of epidemiological and ecological information collected from field studies that sample hosts and their ticks, including tick clustering by host, tick infection status, and the life stage of individual ticks, while inferring whether hosts are infected and infectious as a latent variable. By integrating these components, the framework links infection risks for ticks via specific transmission pathways, facilitating the estimation of pathogen circulation levels in both hosts and ticks, as well as the probability of transmission via each pathway. Furthermore, the framework allows for the incorporation of both tick- and host-level variables to assess their effects on transmission probability, providing a robust and flexible tool for understanding pathogen ecology.

We first describe the model within this framework and then evaluate its ability to recover true parameter values using simulation-based model assessments. Specifically, we assess the impact of sample size (defined as the number of hosts from which ticks are collected) and assumptions regarding cofeeding transmission on parameter estimation, to provide guidance for future sampling strategies to apply this framework.

Next, as a case study, by fitting the model to field data collected in Forêt Domaniale de Haye region in Northeast France in 2023, we estimate parameters for the transmission of *Borrelia garinii*, *Borrelia valaisiana*, and *Anaplasma phagocytophilum* at the bird-tick interface, focusing on the tick *Ixodes ricinus* and three bird species: *Turdus merula* (common blackbird), *T. philomelos* (song thrush) and *Erithacus rubecula* (European robin). *I. ricinus* is the most widespread tick species in Europe and transmits pathogens that pose significant risks to both public and animal health, making it a natural choice for this case study. Among the pathogens it transmits, we selected *B. garinii* and *B. valaisiana* because both are commonly associated with birds in Europe [18,19], yet they may differ in key transmission dynamics. Both *B. garinii* and *B. valaisiana* can cause Lyme disease in humans, with *B. garinii* being the second most common causative agent of human Lyme disease in Europe [18]. *A. phagocytophilum* was chosen as a contrasting bacterial pathogen with potentially different transmission ecology. *A. phagocytophilum* causes human granulocytic anaplasmosis (HGA), although it is reported less frequently in Europe than in the United States, where most HGA cases have been reported globally [20,21]. Notably, while the *Borrelia* genospecies are spirochetes that typically reside extracellularly in host tissues, *A. phagocytophilum* is an obligate intracellular bacterium that lives inside white blood cells. Thus, including two *Borrelia* species alongside *A. phagocytophilum* enables a within-system comparison to evaluate whether the framework can resolve pathogen-specific dynamics. The three bird species are selected because they are among the most abundant local passerines, commonly host *I. ricinus* infestations, and are frequently infected by all three pathogens. In addition, we have external datasets from

the same region with which to evaluate the reliability of the model estimates within this bird-tick system. However, it should be noted that our modelling framework is also readily applicable to other tick-host systems.

Finally, we further validate the model's reliability by comparing its estimates of pathogen circulation levels in ticks or birds with corresponding values from external datasets collected from the same area, which are not used during model fitting.

## Materials and methods

### Model overview

Our model aims at estimating the prevalence of infection at both host and tick levels, as well as the probability of pathogen transmission through direct host-to-tick transmission and indirect cofeeding interactions among ticks. Our model incorporates hierarchical structure to capture the nested nature of host-tick relationships, assuming that ticks share environmental exposure and immune responses on the same hosts. It also accounts for variations at the host level to capture the unobserved heterogeneity between birds in transmission risk to ticks.

Hereafter, our model assumes a 'bird-tick interface', with hosts designated as birds, because we use simulated and field data from this ecosystem as a case study (see 'Simulation-based model assessments' and 'Model fitting to bird-tick pathogen data', respectively). Importantly, for model fitting, we use datasets routinely collected as part of field surveys studying bird-tick-pathogen interactions, including the infection status and life stage (i.e., larva or nymph) of individual ticks, as well as the identity of the birds from which these ticks were collected (i.e., host membership). For tick life stages, our model considers larvae and nymphs but not adults, because they are rarely found on birds, as observed in our field survey datasets. Furthermore, additional details about individual hosts (e.g., host species and age) and ticks (e.g., engorgement level and cofeeding ticks) would enable statistical investigations into specific factors associated with host-to-tick transmission and cofeeding transmission, as illustrated with our bird-tick pathogen data (See 'Model fitting to bird-tick pathogen data').

Input data and parameters estimated are provided in Table 1. The minimum data requirements include, for each bird sampled, (i) the number of ticks identified and (ii) the number of ticks sampled and tested, and for each tick sampled and tested, its (a) life stage, (b) pathogen test results, and (c) the id of the bird from which it was sampled. Other data on other bird- and tick-level variables can be used to assess their impact on the respective transmission probabilities described below.

### Model structure

For tick $i$, the overall infection probability incorporates both bird-to-tick and cofeeding transmission routes, in addition to the probability that the tick was already infected before feeding on the bird from which it was collected ($P_L$ for larvae and $P_N$ for nymphs) through transovarian and/or transstadial transmission. In order words, the model assumes that, if pathogen transmission does not occur, the infection probability for tick $i$ defaults to $P_L$ or $P_N$. However, if pathogen transmission occurs, the infection probability depends on the force of infection arising from bird-to-tick, cofeeding, or both transmission routes. It should be noted that bird-to-tick and cofeeding transmission are respectively modelled as sequences of conditional probabilities, corresponding to distinct epidemiological events, as defined below.

For tick $i$, its infection probability is modelled as follows for each transmission pathway:

#### 1. Bird-to-tick transmission probability

As the first condition for bird-to-tick transmission, a bird needs to be infected and infectious at the time of feeding. For bird $k$ from which tick $i$ originates $k(i)$, its probability of being infectious, $p_B\left[k(i)\right]$, is modelled as a logistic function of the bird's characteristics:

$$p_B\left[k(i)\right] = \text{ilogit}(\alpha_{p_B} + X_B[k(i)] \cdot \beta_{p_B}) \tag{1}$$

**Table 1. Input data and parameters estimated for the bird-tick system.**

| Input data | Definition | | |
|---|---|---|---|
| **Bird level** | | | |
| Bird species | *Turdus merula* (common blackbird), *T. philomelos* (song thrush), or *Erithacus rubecula* (European robin) | | |
| Bird age class | Young or adult | | |
| No. of tested ticks | The number of ticks sampled from the bird and tested | | |
| No. of identified ticks | The number of ticks identified from the bird | | |
| **Tick level** | | | |
| Tick life stage | Larva or nymph | | |
| Tick engorgement level | Mild, moderate, or full | | |
| Tick infection status | PCR positive or negative to *Borrelia. garinii*, *B. valaisiana*, or *Anaplasma. Phagocytophilum* | | |
| Bird id | The id of the bird from which the tick was sampled | | |
| **Parameters** | **Modelled as[a]:** | **Prior[b]** | **Equations** |
| Probability of a bird being infectious ($p_B$) | A linear combination of bird-level coefficients ($\beta_{p_B}$): Intercept Bird species (ref: *T. merula*) Age group (ref: old) | Normal(0,1) | Eqs. 1 and 2 |
| Probability of bird-to-tick transmission ($p_{B \to T}$) | A linear combination of tick-level coefficients ($\beta_{p_{B \to T}}$): Intercept Engorgement level (ref: mild) Life stage (ref: larvae) | | Eq. 3 and 4 |
| Probability of cofeeding transmission ($p_{T \to T}$) | A linear combination of tick-level coefficients ($\beta_{p_{T \to T}}$): Intercept Engorgement level (ref: mild) Life stage (ref: larvae) | | Eq. 8 |
| Probability of a nymph being infected before feeding ($p_N$) | A single parameter $p_N$ | Beta [1] | Eqs. 11 and 12 |
| Offset parameter for the probability of a larvae being infected before feeding ($\delta$) | $p_L < p_N$ $p_L = ilogit(logit\,(p_N) - \delta)$ | Normal(0,1)T[0,] | |
| Standard deviation for bird-level random effect ($\sigma$) | A single parameter $\sigma$ | Uniform(0,10) | Eq. 10 |
| Covariance for bird-to-tick and cofeeding transmission ($\rho$) | A single parameter $\rho$ | Normal(0,1) | |

When a parameter was defined as a linear combination of coefficients, (a) "ref" indicates the reference group and (b) Normal(0,1) was used as a prior for each coefficient.

$\alpha_{p_B}$ is an intercept representing the baseline probability of a bird being infectious, while $X_B[k(i)]$ is a vector of bird-specific characteristics with $\beta_{p_B}$ being a vector of coefficients for these characteristics (i.e., bird species and age group). Whether bird $k(i)$ is infectious or not, $INF_B\left[k(i)\right]$, is drawn from a Bernoulli distribution:

$$INF_B\left[k(i)\right] \sim \text{Bernoulli}(p_B\left[k(i)\right]) \qquad (2)$$

If bird $k(i)$ is determined to be infectious, the probability of direct bird-to-tick transmission (i.e., the probability of an effective bird-to-tick contact), $p_{B \to T}[i]$, becomes applicable for each individual tick $i$ attached to that bird. $p_{B \to T}[i]$ is modelled as a logistic function of the tick's characteristics.

$$p_{B \to T}[i] = ilogit(\alpha_{p_{B \to T}} + X_T[i] \cdot \beta_{p_{B \to T}}) \qquad (3)$$

$\alpha_{p_{B \to T}}$ is an intercept representing the baseline probability of an effective contact with an infectious bird, leading to bird-to-tick transmission. $X_T[i]$ is a vector of tick-specific characteristics with $\beta_{p_{B \to T}}$ being a vector of coefficients for these

characteristics (i.e., tick life stage and engorgement level). As a result, the overall bird-to-tick transmission probability for tick $i$ collected from bird $k(i)$ becomes $INF_B\left[k(i)\right] \cdot p_{B \to T}[i]$.

It should be noted that the framework does not consider the transmission from a tick to its host sampled together. Although this pathway is important, we hypothesise that the host must have been infected prior to being fed on by the sampled ticks. This is because the short feeding period of the tick would not provide sufficient time for the host to become infectious, while localised cofeeding transmission would remain possible [11,22].

### 2. Cofeeding transmission probability

We assume that cofeeding transmission involves two conditional probabilistic steps. As the first condition, a tick must be feeding on the same bird with at least one infected nymph, and then as the second condition, cofeeding transmission then must occur from infected nymph(s) to non-infected tick(s) with the probability, $p_{T \to T}$ (i.e., the probability of an effective contact between cofeeding ticks).

Ideally, knowing the first condition would require detailed information on the infection status of all individual ticks sampled from the same bird. Although in practice this is not feasible, in most sampling situations, ticks are collected from easily accessible areas on birds, such as on the head around the eyes and beak, where ticks are more visible and often cofeed in groups. The infection status of these collected ticks can, therefore, serve as an approximate indicator of whether cofeeding transmission could occur among them. Following this logic, when fitting the model to the bird-tick pathogen data, we assume that cofeeding transmission occurs if there is at least one infected nymph among those feeding on the same bird, and $p_{T \to T}$ is estimated separately (see 'Model fitting to bird-tick pathogen data' below).

However, this approach complicates simulation-based model assessment, as simulating the infection status of individual ticks requires information on the presence of infected nymphs feeding on the same bird, leading to a circular dependency in the data structure. Therefore, for general model assessment purposes, we propose two alternative approaches. In the first approach, the probability that tick $i$ is feeding with infected nymphs depends on the estimated number of ticks on its bird $k(i)$ at the time of collection. That is, it assumes that the probability increases with the number of ticks feeding on the bird (Cofeeding probability assumption 1). We quantify this relationship by fitting a logistic regression model that assesses the association between the presence of at least one infected nymph and the estimated number of ticks in the bird-tick pathogen data. We use the predicted values from this logistic regression model as the probability parameter in a Bernoulli distribution, $p_{cofeeding_1}\left[k(i)\right]$. This allows us to estimate whether tick $i$ and other ticks feeding on bird $k(i)$ are in proximity to infected nymphs, $Cofeeding_1[i, k]$:

$$p_{cofeeding_1}\left[k(i)\right] = \frac{1}{1 + e^{-(\beta_0 + \beta_1 X[k(i)])}} \tag{4}$$

$$Cofeeding_1\left[i, k(i)\right] \sim \text{Bernoulli}(p_{cofeeding_1}\left[k(i)\right]) \tag{5}$$

In the second approach, we assume that the probability that tick $i$ is feeding with infected nymphs is defined by the proportion of hosts with at least one infected nymph, using a Bernoulli distribution (Cofeeding probability assumption 2), and we define $p_{cofeeding_2}$:

$$p_{cofeeding_2} = \frac{\text{No. of sampled hosts with at least one infected nymph}}{\text{Total no. of sampled hosts}} \tag{6}$$

$$Cofeeding_2\left[i, k(i)\right] \sim \text{Bernoulli}(p_{cofeeding_2}) \tag{7}$$

The implications of these alternative approaches are tested and discussed through simulation-based model assessment (see 'Simulation-based model assessments' below).

   Finally, if tick $i$ is determined to be feeding with infected nymphs based on either of the above approaches, the probability of cofeeding transmission, $p_{T \to T}[i]$, is applicable, modelled as a logistic function of the tick's characteristics.

$$p_{T \to T}[i] = \text{ilogit}(\alpha_{p_{T \to T}} + X_T[i] \cdot \beta_{p_{T \to T}}) \tag{8}$$

$\alpha_{p_{T \to T}}$ is an intercept representing a baseline cofeeding transmission probability, while $X_T[i]$ is a vector of tick-specific characteristics with $\beta_{p_{T \to T}}$ being a vector of coefficients for these characteristics (i.e., tick life stage and engorgement level). As a result, the overall cofeeding transmission probability for tick $i$ collected from host $k(i)$ becomes $Cofeeding\left[i, k(i)\right] \cdot p_{T \to T}[i]$.

### 3. Combined force of infection on ticks while attached on the sampled host

We model the combined force of infection as follows:

$$FOI_1\left[i, k(i)\right] = 1 - \left(1 - INF_B\left[k(i)\right] \cdot p_{B \to T}[i]\right) \cdot \left(1 - Cofeeding\left[i, k(i)\right] \cdot p_{T \to T}[i]\right) \tag{9}$$

$$FOI_2\left[i, k(i)\right] = \text{ilogit}(\text{logit}\left(FOI_1\left[i, k(i)\right]\right) + \rho \cdot INF_B\left[k(i)\right] \cdot Cofeeding\left[i, k(i)\right] + \eta\left[k(i)\right]) \tag{10}$$

$FOI_2\left[i, k(i)\right]$ represents the probability that tick $i$ on bird $k(i)$ becomes infected through either bird-to-tick transmission, cofeeding transmission, or both, accounting for potential non-independence between the two transmission pathways and heterogeneity in transmissibility at the bird level with $\rho$ and $\eta\left[k(i)\right]$, respectively, while $FOI_1\left[i, k(i)\right]$ does not account for these factors. $\rho$ represents a covariance effect between host-to-tick transmission and cofeeding transmission. That is, these two transmission routes are assumed not independent. $\eta\left[k(i)\right]$ represents a random effect for each host $k(i)$, accounting for unobserved variability in transmissibility among hosts. $\eta\left[k(i)\right]$ was sampled from a Normal distribution with the mean of 0 and the standard deviation, σ, estimated through model fitting.

### 4. Adjusted infection probability, accounting for ticks being infected prior to feeding the sampled host

Finally, the combined force of infection, $FOI_2\left[i, k(i)\right]$, is adjusted to account for the probability that tick $i$ is already infected prior to feeding on host $k$, $p_{s(i)}$:

$$FOI_3\left[i, k(i)\right] = \left(1 - p_{s(i)}\right) \cdot FOI_2\left[i, k(i)\right] + p_{s(i)} \tag{11}$$

$$p_L = \text{ilogit}(\text{logit}\left(p_N\right) - \delta) \tag{12}$$

The prior-infection probability is modelled separately for larvae ($s(i) = L$) and nymphs ($s(i) = N$). For example, if prior infection is assumed only for nymphs, $FOI_3\left[i, k(i)\right]$ in Eq. 11 becomes $FOI_2\left[i, k(i)\right]$ for larvae and remains $\left(1 - p_N\right) \cdot FOI_2\left[i, k(i)\right] + p_N$ for nymphs. $\delta > 0$ is an offset parameter ensuring $p_L < p_N$. That is, nymphs are assumed to have a higher probability than larvae, considering that they have a blood meal at their larval stage, whereas larvae can only be infected prior to feeding through transovarial transmission. Previously interrupted blood meals are also possible, but nymphs would still, on average, have more opportunities for such events and therefore a higher prior-infection probability than larvae.

Finally, the infection status of tick $i$ from bird $k(i)$ is drawn from a Bernoulli distribution, with $FOI_3[i, k]$ as the probability parameter:

$$INF_T[i, k(i)] \sim \text{Bernoulli}(FOI_3[i, k(i)])$$ (13)

Therefore, the log-likelihood of the observed tick infection data, given the fixed parameters $\theta$ (Table 1), the random effects $\eta$, and the latent bird infection status $INF_B$, becomes:

$$\log p(y | \theta, \eta, INF_B) = \sum_{i=1}^{T} [y_i \log(FOI_3[i, k(i)]) + (1 - y_i) \log(1 - FOI_3[i, k(i)])]$$ (14)

## Simulation-based model assessments

Simulation-based model assessments were conducted by fitting the model to synthetic datasets that mimic the structure of the empirical bird-tick pathogen data used in this study. We evaluated three aspects: (i) the model's ability to recover true parameter values, (ii) the impact of sample size (number of sampled birds, and the resulting number of ticks sampled), and (iii) the effect of how ticks are distributed across birds—and how this distribution relates to the probability of cofeeding with infected nymphs—on estimation performance.

### 1. Parameter identifiability

The first assessment focused on the model's ability to recover true parameter values by fitting it to synthetic datasets and comparing the posterior parameter estimates with the true parameter values used to generate those datasets. We generated 100 synthetic datasets. The number of birds was assumed to be the same as the number of birds in the bird-tick pathogen empirical data (see "Model fitting to bird-tick pathogen data" below). For each bird, the number of ticks was simulated using a negative binomial distribution, with the mean ($\mu = 7.130$), overdispersion ($k = 1.217$), and larvae-to-nymph ratio (0.472) informed by the bird-tick pathogen data. Other parameter values were randomly selected from the joint posterior distribution of the baseline model fitted to the *B. garinii* dataset among the bird-tick pathogen data. Based on the parameter values selected, the infection status of each tick was then determined by $FOI_3[i, k]$ in Eq. 13. Note that we did not draw parameters values independently and directly from priors, because doing so can produce unrealistic combinations of parameter values and infection scenarios; instead, drawing from the joint posterior allows parameter recovery to be evaluated within empirically plausible ranges while preserving posterior dependencies among parameters. We used the *B. garinii* posterior because its underlying dataset showed a significant association between the presence of at least one infected nymph and the estimated number of ticks in the bird-tick pathogen data, enabling us to assess parameter recovery under the more complex model assumption (Cofeeding probability assumption 1, $p_{cofeeding_1}$ in Eq. 4). The impact of assuming a constant cofeeding probability (Cofeeding probability assumption 2, $p_{cofeeding_2}$ in Eq. 6) was evaluated in the subsequent assessment steps.

The model was subsequently fitted to each synthetic dataset using Gibbs Markov chain Monte Carlo (MCMC) sampling algorithm in JAGS [26] in R 4.3.2. Across the model fits, the following model performance metrics were obtained for each parameter: (i) the distribution of bias, defined as the difference between the median posterior estimate and the true parameter value, (ii) 95% highest density intervals (HDI) coverage, defined as the percentage of the synthetic datasets in which the true parameter value lies within the 95% HDI, and (iii) the distribution of precision, defined as the inverse of the 95% HDI width.

### 2. Impact of sample size on parameter estimation

The second assessment examined the impact of sample size on the model performance metrics. In most sampling situations where hosts are captured first and ticks are collected from these hosts, it is realistic to treat the number of hosts

sampled as the primary design target because the number of ticks collected depends primarily on the number of hosts captured. Accordingly, we defined sample size as the number of birds sampled, with the number of ticks subsequently determined by the assumed per-bird tick-burden distribution (see the next section, "Impact of tick overdispersion on parameter estimation"). We then assessed the effect of sample size on model estimates across bird sample sizes ranging from 10 to 250, using 100 synthetic datasets for each sample size.

For the same reason as in the assessment on parameter identifiability, we generated synthetic datasets using posterior parameter estimates from the model fitted to the *B. garinii* dataset (see "Model fitting to bird-tick pathogen data" below), rather than relying on random samples from weakly informed prior distributions.

### 3. Impact of tick overdispersion on parameter estimation

Finally, we examined how heterogeneity in tick burdens across birds affects estimation performance, because it changes both the amount and distribution of information in the data and can therefore influence model fitting. To do this, we fitted the model to synthetic datasets generated under different scenarios for how ticks are distributed across birds: (i) overdispersed tick burdens following a negative binomial distribution, as observed in the bird-tick pathogen data (baseline assumption, $\mu = 7.130$, overdispersion parameter $k = 1.217$); (ii) increased overdispersion, also under a negative binomial distribution ($\mu = 7.130$, $k = 0.561$); and (iii) non-overdispersed tick burdens following a Poisson distribution ($\mu = 7.130$). Each scenario was combined with the two assumptions about the probability of cofeeding with infected nymphs described above: constant across birds or varying with observed tick burden.

## Model fitting to bird-tick pathogen data

### 1. Bird-tick pathogen data

The bird-tick pathogen data included the infection and demographic information of 187 larvae and 209 nymphs as well as 26 blackbirds (*Turdus merula*), 30 song thrushes (*Turdus philomelos*), and 12 robins (*Erithacus rubecula*) from which these ticks were collected. The birds and ticks were sampled in the Forêt Domaniale de Haye (48.6779°N, 06.1001°E), Northeast France, from June 14 to July 4 in 2023. Bird capture and blood sampling were conducted under licence from the Centre de Recherches sur la Biologie des Populations d'Oiseaux (CRBPO, BF permit no. 1073). Bird hosts were captured using mist nets. *T. merula* and *T. philomelos* in the *Turdidae* family, and *E. rubecula* in the *Muscicapidae* family, were selected for sampling, given that they are known to exhibit the highest tick prevalence among common forest-dwelling passerines [9]. Each captured bird was checked visually for ticks, with particular attention on the head, where ticks are most found [23]. The number of ticks detected on each bird was recorded, and ticks were removed using tick-removal tools or fine pointed forceps, and then immediately placed in microtubes filled with 70% ethanol for later laboratory analysis. When tick removal was too tricky, ticks were not collected to avoid injuring the birds. In cases where more than 10 ticks were detected on a bird, only 10 ticks were collected and analysed for pathogen presence; engorged ticks were prioritised to maximise the probability of pathogen detection within the available budget. This resulted in testing not all the ticks identified in 35.3% of the captured birds. From each captured *T. merula* bird, 300–400μl of blood was drawn from the brachial vein using sterile 27G needles, collected into 100μl heparinised tubes, and deposited in duplicate onto FTA cards for laboratory analysis. Birds were released after confirming that bleeding had stopped.

For each tick, the infection status of *B. garinii*, *B. valaisiana*, and *A. phagocytophilum* was determined using high-throughput microfluidic real-time PCR, performed on a BioMar real-time PCR system (Standard BioTools, USA). Primers and probes sequences are detailed in Melis, Batisti Biffignandi [24]. This technique uses the 48.48 Dynamic Array (Standard BioTools, USA), which features two sets of 48 inlets, allowing simultaneous loading of 48 samples and 48 primers and probe (assay) mixtures. All 2,304 sample-assay combinations are processed and tested simultaneously by the automated microfluidic system, requiring only few total microliters of each sample and assay. The results were

confirmed for each pathogen by conducting conventional PCR on around 20% of the positive samples; all showed >99% sequence homology at the species level (see Melis, Batisti Biffignandi [24] and Michelet, Delannoy [25] for details).

The life stage (larva or nymph) and engorgement level (mild, moderate, or full) of the ticks, and the id of the bird from which it was collected were recorded.

Additionally, prevalence estimates of *B. garinii*, *B. valaisiana*, and *A. phagocytophilum* in questing nymphs collected from the same Forêt Domaniale de Haye area in late June to early July from 2021 to 2023 were used to evaluate whether the model's estimated probability of a nymph having been infected prior to feeding, $p_{T_{inf}}[i]$, was comparable to general infection levels in questing nymphs. Here, questing nymphs were collected by dragging a cotton sheet over 200m² area in two different locations within the forest and tested by PCR as done for ticks collected from birds. For each bird, its species (*T. merula*, *T. philomelos*, and *E. rubecula*) and age (≤1-year-old or >1-year-old) were recorded. The molecular testing results for *A. phagocytophilum* were also available for the blood of birds from which ticks were collected. However, this information was not used for the model fitting itself but only to assess whether the model's estimated probability of a bird being infected, $p_B[k(i)]$, was comparable to actual molecular testing result.

## 2. Model fitting to observed bird-tick pathogen data

All models were fitted to bird-tick *B. garinii*, *B. valaisiana*, and *A. phagocytophilum* data separately using Gibbs MCMC sampling algorithm in JAGS [26] in R 4.3.2, with random initial values given to four chains. After discarding the first 30,000 posterior samples as burn-in, another 30,000 were retained per chain, and convergence was assessed by visual inspection of MCMC trace plots and Gelman-Rubin convergence diagnostic (<1.01) [27].

In the full model, (i) the probability of a bird being infectious ($p_B$), (ii) the probability of direct bird-to-tick transmission ($p_{B \to T}$), and (iii) the probability of cofeeding transmission ($p_{T \to T}$) were modelled as logistic functions of all available bird- and tick-level information. Bird species and bird age were included as predictors for $p_B$, while tick life stage and engorgement level were used as predictors for $p_{B \to T}$ and $p_{T \to T}$. Importantly, $p_{T \to T}$ was valid for tick $i$ on bird $k(i)$ only when there was at least one infected nymph among those sampled from the same bird, i.e., $Cofeeding[i, k(i)] = 1$ in $Cofeeding[i, k(i)] \cdot p_{T \to T}[i]$ for the overall cofeeding transmission probability.

After model fitting, a posterior predictive check was performed using the proportion of ticks that tested positive to each pathogen, differentiated by bird species and tick life stage, as the predicted measure. For each predictor, its effect size was evaluated relative to a reference group, and its statistical significance was assessed by comparing model fits between the full model and the model without the predictor based on deviance information criterion (DIC), obtained with the *dic.sample* fuction of the *rjags* package [26]. A decrease in DIC greater than 5 units was considered to indicate a statistically significantly better model fit.

## Results

### 1. Results of simulation-based model assessment

The comparison between posterior parameter estimates and true parameter values indicated that the model could recover true parameter values, with no systematic bias observed in posterior parameter estimates and acceptable 95% highest density intervals (HDI) coverages ranging between 93.0% and 97.0% (Fig A in S1 Text).

Under various scenarios for the probability of cofeeding with infected nymphs and tick distributions across hosts, 95%HDI coverages remained robust across all sample sizes, with approximately 95% of true parameter values falling within these intervals across simulations (Fig B in S1 Text). Precision and bias improved noticeably with larger sample sizes when the number of ticks per bird was overdispersed following a negative binomial distribution (overdispersion parameter k = 1.217 or 0.561), compared to a Poisson distribution, and when cofeeding probability was allowed to vary with tick numbers ($p_{cofeeding_1}$, blue circles and diamonds, Figs 1 and 2). In contrast, assuming a constant cofeeding

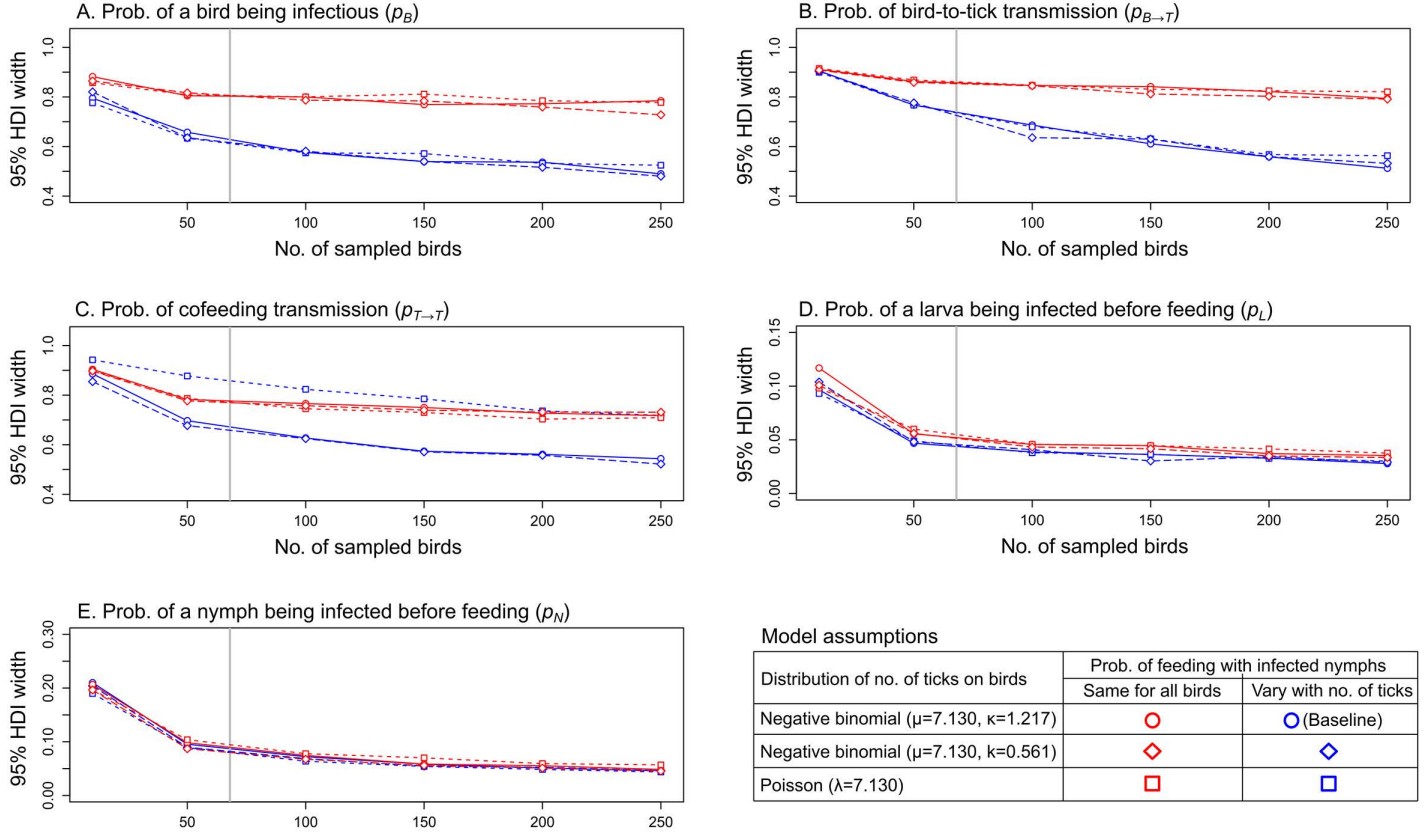

**Fig 1. Precision of posterior parameter estimates across different sample sizes and model assumptions.** Point shapes indicate precision under assumptions about the distribution of tick numbers on birds, while point colours indicate precision under assumptions about the probability of cofeeding with infected nymphs ($p_{cofeeding_1}$: blue points; $p_{cofeeding_2}$: red points). Blue circles represent precision under the baseline assumptions. Vertical lines the number of birds in the bird-tick pathogen data.

probability across ticks led to consistently poor precision for parameters related to bird-to-tick and cofeeding transmission, with minimal improvement observed with larger sample sizes ($p_{cofeeding_2}$, red points, Fig 1A, 1B, and 1C). Similarly, assuming a non-overdispersed (i.e., Poisson) tick distribution, even with $p_{cofeeding_1}$, resulted in a poor precision for the probability of cofeeding transmission ($p_{T \rightarrow T}$) across all sample sizes (blue squares, Fig 1C). These trends in precision were also reflected in bias for $p_{T \rightarrow T}$ and for the covariance parameter ($\rho$), which captures the relationship between host-to-tick and cofeeding transmission (blue squares, Fig 2C and 2G).

## 2. Results of model fitting to observed bird-tick pathogen data

For all three pathogens studied (*B. garinii*, *B. valaisiana*, and *A. phagocytophilum*), the full model accurately predicted the empirical proportion of infected ticks by bird species and tick life stage (Fig 3). The full model provided a significantly better fit to the data compared to the baseline model, which assumed no effect of bird- and tick-level predictors (Table 2). Also, the model assuming cofeeding transmission yielded a significantly better fit to the data than the model assuming no cofeeding transmission (vs. Model 1.7, Table 2). For *A. phagocytophilum*, incorporating engorgement level into the probability of bird-to-tick transmission also improved the model fit (vs. Model 1.4, Table 2), whereas incorporating engorgement level into the probability of cofeeding transmission did not (vs. Model 1.6, Table 2). Other assumptions did not explain the

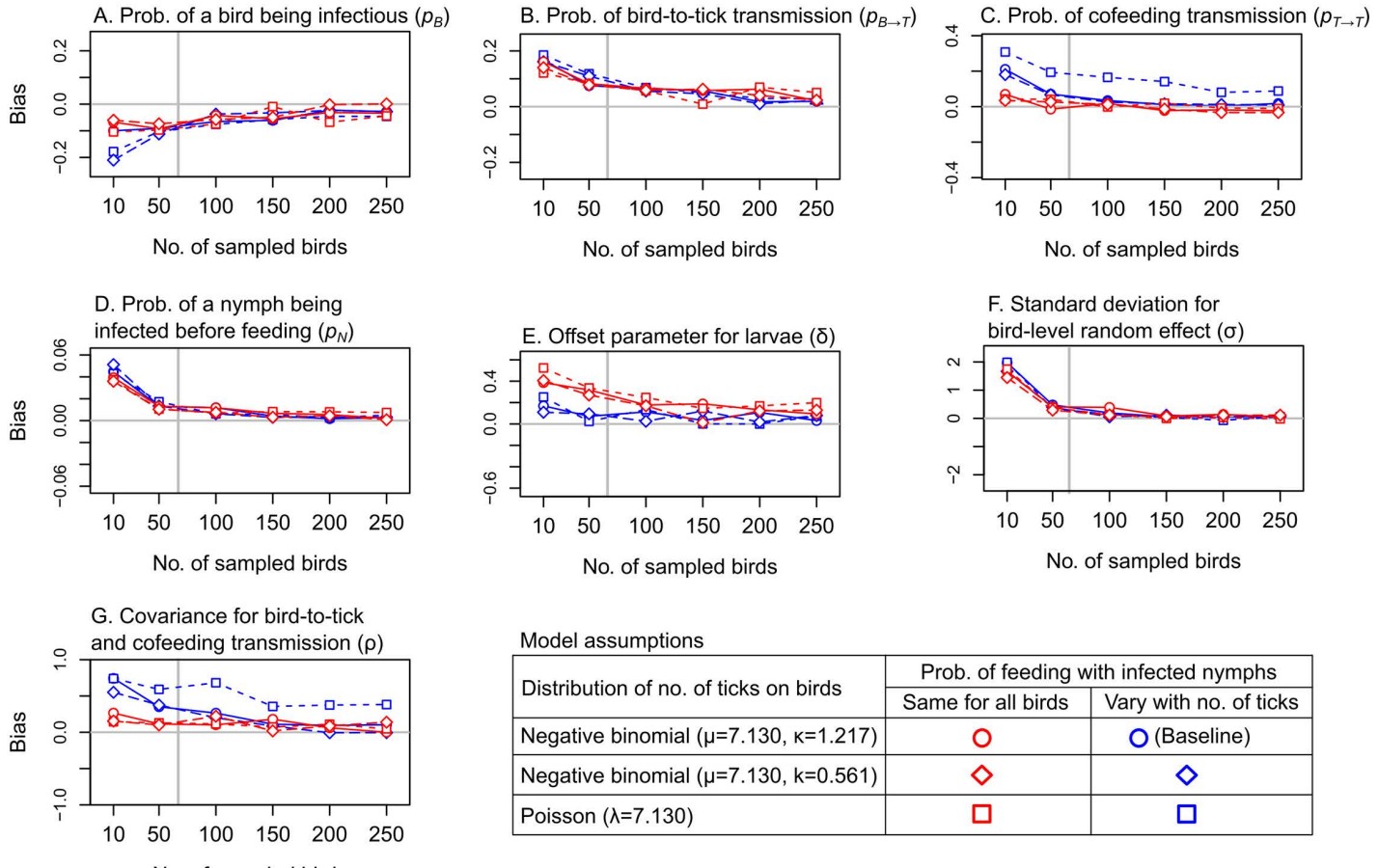

**Fig 2. Bias of posterior parameter estimates across different sample sizes and model assumptions.** Point shapes indicate bias under assumptions about the distribution of tick numbers on birds, while point colours indicate bias under assumptions about the probability of cofeeding with infected nymphs ($p_{cofeeding_1}$: blue points; $p_{cofeeding_2}$: red points). Blue circles represent bias under the baseline assumptions. Vertical lines the number of birds in the bird-tick pathogen data.

data significantly better than the full model (DIC < 5, Table 2). However, to facilitate comparison across the three pathogens and to present the full set of transmission pathways that can be considered for tick-borne disease transmission in a host-tick system, the results presented in this study are based on the full model.

For all three pathogens, the median posterior probability of a bird being infectious ranged between 0.06 and 0.35 across different bird species and age groups, although no significant differences were observed between these groups (Figs 4A, 5A, and 6A). For *A. phagocytophilum*, the posterior probability of a *T. merula* bird being infectious was estimated to closely match the proportion of the *T. merula* birds that tested positive for this pathogen, which was used as a value only for comparison but not used for model fitting (Fig 6A).

The median posterior probability of a tick becoming infected from an infectious bird was highest for fully engorged ticks, particularly for *A. phagocytophilum* (0.66, 95%HDI: 0.22 to 0.95), compared to *B. valaisiana* (0.21, 95%HDI: 0.01 to 0.74) and *B. garinii* (0.48, 95%HDI: 0.10 to 0.86) (Figs 4B, 5B, and 6B). Notably, the association between engorgement level and infection probability was particularly strong for *A. phagocytophilum*. Fully engorged ticks were estimated to have 9.67

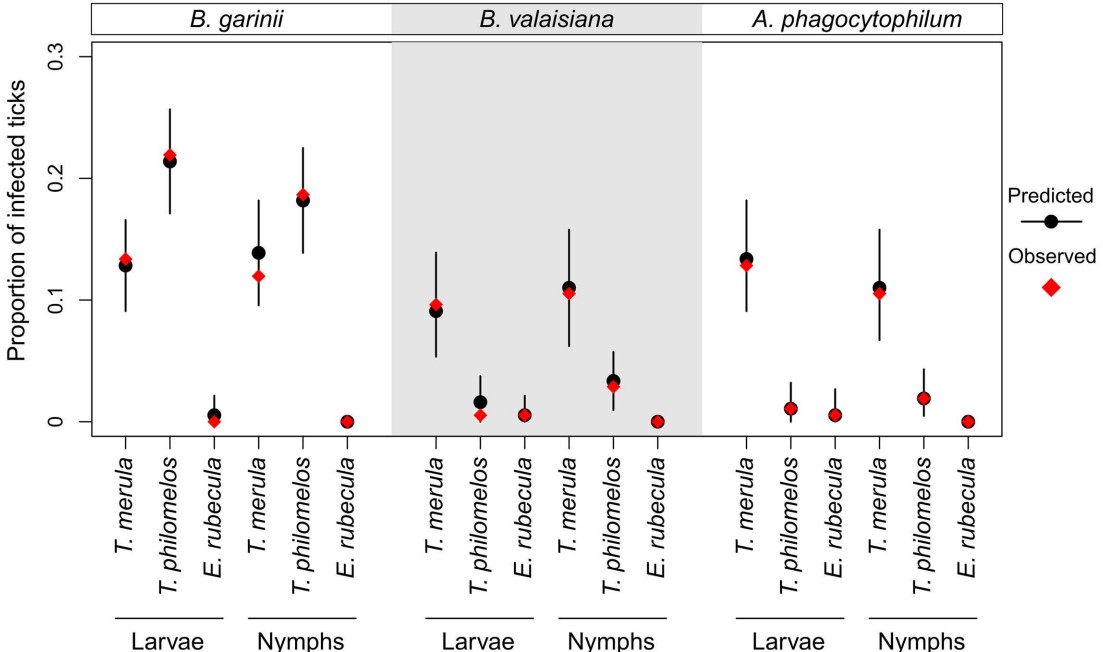

**Fig 3. Full model fit.** For each pathogen, the predicted distribution of the proportion of infected ticks was compared with the observed proportion by bird species and tick life stage. Circles and vertical lines represent the median and 95% percentile intervals of the predicted proportions, while diamonds indicate the observed proportions.

(95%HDI: 1.52 to 53.90) times the odds of becoming infected with *A. phagocytophilum* from an infectious bird compared to mildly engorged ticks (Set 2 in Figs C and D in S1 Text).

The median posterior probability of cofeeding transmission in the presence of at least one infected nymph was estimated to be higher for *B. garinii* (0.58 to 0.73, Fig 4C) than *B. valaisiana* (0.17 to 0.40, Fig 5C) and A. *phagocytophilum* (0.24 to 0.58, Fig 6C) across ticks at different life stages and varying engorgement levels. While cofeeding transmission probability tended to increase with higher engorgement levels, these trends were not statistically significant for all three pathogens (Set 3 in Figs C and D in S1 Text).

The median posterior probability of a tick being already infected prior to feeding the sampled host was estimated to be the highest for *B. garinii* (0.014 for larvae and 0.022 for nymphs, Fig 4D), with *B. valaisiana* (0.002 for larvae and 0.005 for nymphs, Fig 5D) and A. *phagocytophilum* (0.002 for larvae and 0.005 for nymphs, Fig 6D) having similar levels. The probability was estimated to be lower in larvae than in nymphs, with no significant difference observed for a model that did not assume prior infection at the larval stage (Model 1.8, no $p_L$ in Eq. 11) compared with the full model that assumed it ($p_L$ in Eq. 11) (Table 2). Finally, for all three pathogens, the probability of a nymph being infected prior to feeding the sampled host was estimated to fall within the same range observed from empirical data collected independently from the present study in the same forest region in northeast France (red triangles and vertical dashed lines in Figs 4D, 5D, and 6D).

## Discussion

The transmission dynamics of tick-borne pathogens are governed by a complex interplay of factors operating across different layers of tick-host-pathogen interactions [1,28]. At the macro level, the presence of host-to-tick and cofeeding transmission, as well as the ability of ticks to acquire infections prior to a current life stage, add to this complexity. Therefore,

**Table 2. Model comparison.**

| Model | Model assumptions on infection scenarios | | | | Model fits[a] | | | | | |
|---|---|---|---|---|---|---|---|---|---|---|
| | **Prob. of a bird being infectious** | **Prob. of bird-to-tick transmission** | **Prob. of cofeeding transmission** | **Prob. of tick being infected prior to feeding** | *Borrelia garinii* | | *Borrelia valaisiana* | | *Anaplasma phagocytophilum* | |
| | | | | | DIC | ΔDIC | DIC | ΔDIC | DIC | ΔDIC |
| Full model | Bird species Bird age | Tick life stage Engorgement level | Tick life stage Engorgement level | For both larvae & nymphs | 294.3 | 0.0 | 215.4 | 0.0 | 189.7 | 0.0 |
| 1.1 | Bird age | Tick life stage Engorgement level | Tick life stage Engorgement level | For both larvae & nymphs | 294.5 | 0.2 | 216.0 | 0.6 | 193.5 | 3.9 |
| 1.2 | Bird species | Tick life stage Engorgement level | Tick life stage Engorgement level | | 294.6 | 0.2 | 214.0 | -1.4 | 188.9 | -0.8 |
| 1.3 | Bird species Bird age | Engorgement level | Tick life stage Engorgement level | | 295.7 | 1.4 | 217.9 | 2.5 | 191.5 | 1.8 |
| 1.4 | Bird species Bird age | Tick life stage | Tick life stage Engorgement level | | 294.2 | -0.1 | 216.3 | 0.9 | 204.0 | **14.3** |
| 1.5 | Bird species Bird age | Tick life stage Engorgement level | Engorgement level | | 292.4 | -2.0 | 215.9 | 0.5 | 191.1 | 1.4 |
| 1.6 | Bird species Bird age | Tick life stage Engorgement level | Tick life stage | | 296.9 | 2.6 | 218.9 | 3.6 | 182.4 | **-7.2** |
| 1.7 | Bird species Bird age | Tick life stage Engorgement level | NO cofeeding transmission | | 310.1 | **15.8** | 244.7 | **29.4** | 220.2 | **30.5** |
| 1.8 | Bird species Bird age | Tick life stage Engorgement level | Tick life stage Engorgement level | ONLY for nymphs | 296.7 | 2.4 | 217.4 | 2.1 | 194.4 | 4.7 |
| Baseline | NO predictor | NO predictor | NO predictor | For both larvae & nymphs | 316.4 | **22.1** | 265.2 | **49.8** | 246.9 | **57.3** |

DIC Deviance information criterion.

[a]DIC values with >5 DIC difference from the full model are highlighted in bold.

conventional approaches, designed to address simpler and more linear transmission routes, for example, surveillance that estimates infection level in ticks or hosts alone, often fall short of capturing these multifaceted dynamics [1,28].

In this regard, our modelling framework provides a foundation for addressing these challenges by integrating epidemiologically and ecologically linked datasets. Our framework explicitly models major transmission pathways through which a tick collected from its host could be determined as infected. Importantly, within this Bayesian inference framework, parameters are estimated not only based on the tick's infection status but also through a simultaneous consideration of the infection statuses of other ticks from the same host, their respective life stages, and the host's infectious status, which is treated as an unobserved latent variable. Furthermore, our assessment through simulation and model fitting to field bird-tick pathogen data highlights the types of data that can substantially enhance the reliability of parameter estimates, offering valuable guidance for the design of future studies aimed at applying this framework.

From a modelling perspective, the ultimate goal in understanding tick-borne disease transmission dynamics is to estimate parameters in host-tick-pathogen systems based on data on host and tick population dynamics and pathogen transmission over time. However, the data required to achieve this are extremely difficult to obtain because of, among many other factors, the involvement of multiple hosts, limitations in quantifying host and tick populations, and the presence of multiple transmission pathways. Key transmission parameters estimated using our cross-sectional framework can therefore complement data required for future longitudinal models. For example, in a given study area, the framework could be applied to individual host-tick systems involved in the transmission of a pathogen of interest, including birds, small mammals, and large mammals, with separate model adaptations depending on the assumed roles of hosts (e.g., amplifying or non-amplifying hosts) and the tick life stages most commonly found on those hosts. The resulting parameter

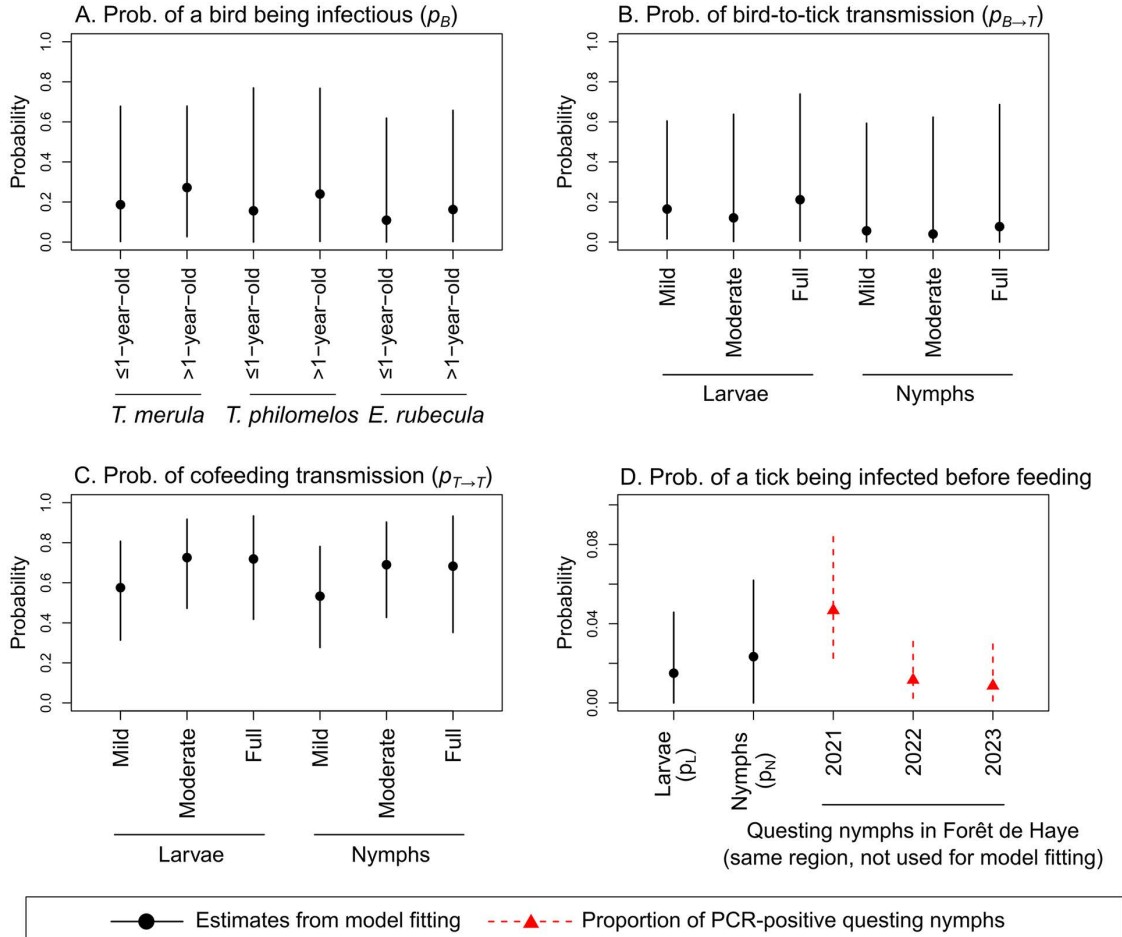

**Fig 4. Posterior parameter distribution from the full model fitted to the *B. garinii* dataset.** Black points and vertical lines represent the posterior median and 95%HDI estimates. Panel A shows the probability that a bird was infected and infectious by bird species and age group. Panel B shows the probability that a tick was infected from an infectious bird, by life stage and engorgement level. Panel C shows the probability that a tick was infected via cofeeding with other infected nymphs, by life stage and engorgement level. Panel D shows the probability that a tick was already infected before feeding, by life stage. Red triangles and dashed lines represent the proportion of questing nymphs that tested positive to *B. garinii* in the same Forêt Domaniale de Haye region in Northeast France, which was not used for model fitting.

estimates could then inform longitudinal models collectively, building upon tick population dynamics models such as Kim, Jaulhac [29], combined with demographic and epidemiological data collected over time in the same area. Although such aims still present substantial logistical and ecological challenges, they are likely more feasible than trying to collect all the data needed to directly parameterise complex models.

Within our framework, each transmission pathway is modelled as a sequence of conditional probabilities, each reflecting a specific epidemiological event. For host-to-tick transmission, our framework distinguishes the probability of a host being infectious from the probability of host-to-tick transmission from an infected host. This differentiation is crucial because host infectiousness does not necessarily result in host-to-tick transmission, influenced by a variety of factors involved in tick-host-pathogen interactions [3]. In particular, for pathogens that survive through tissue colonisation, such as *Borrelia* species, the likelihood of host-to-tick transmission would depend on factors such as the feeding site on the host skin, the concentration of pathogens at that site, and the blood volume ingested by the tick [11,30,31]. Supporting this,

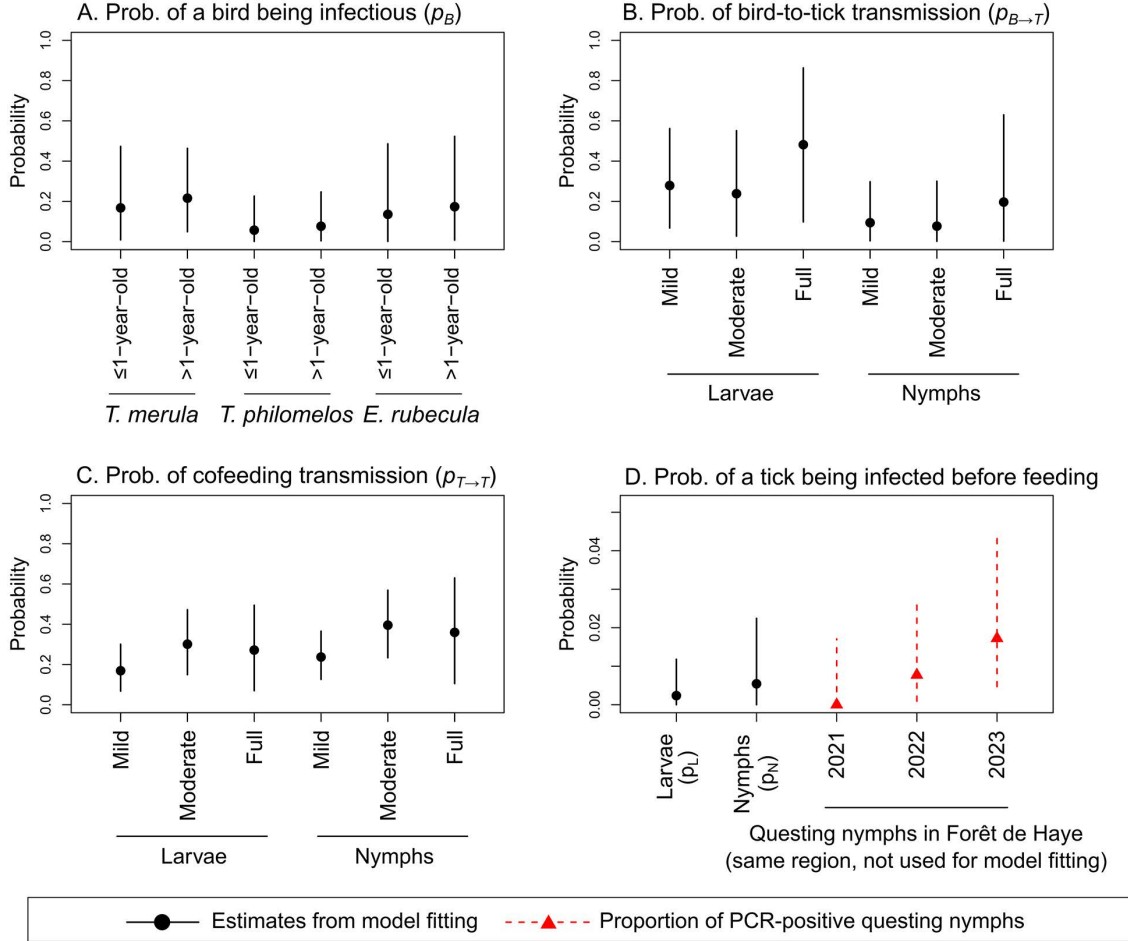

**Fig 5. Posterior parameter distribution from the full model fitted to the *B. valaisiana* dataset.** Black points and vertical lines represent the posterior median and 95%HDI estimates. Panel A shows the probability that a bird was infected and infectious by bird species and age group. Panel B shows the probability that a tick was infected from an infectious bird, by life stage and engorgement level. Panel C shows the probability that a tick was infected via cofeeding with other infected nymphs, by life stage and engorgement level. Panel D shows the probability that a tick was already infected before feeding, by life stage. Red triangles and dashed lines represent the proportion of questing nymphs that tested positive to *B. valaisiana* in the same Forêt Domaniale de Haye region in Northeast France, which was not used for model fitting.

studies on *Borrelia* infections have shown that only a fraction of xenodiagnostic ticks became infected after feeding on the same birds [12,13].

On the other hand, the probability of *A. phagocytophilum* transmission would be less heterogenous across feeding sites compared to that of *Borrelia* transmission, considering that *A. phagocytophilum* survives by preferentially replicating within blood neutrophil granulocytes [6]. That is, the greater the volume of blood a tick consumes, the higher the likelihood that the blood contains the pathogen, potentially with higher pathogen loads, although the impact of other unobserved tick-host-pathogen interactions cannot be ignored. Our results support this hypothesis, showing a significantly higher probability of *A. phagocytophilum* transmission for fully engorged ticks. This pattern was less pronounced with *B. garinii* and *B. valaisiana*.

Our results also suggest that cofeeding transmission is a critical factor for all three pathogens, as incorporating it into the model significantly improved the fit for each pathogen compared to models that excluded this mechanism. This

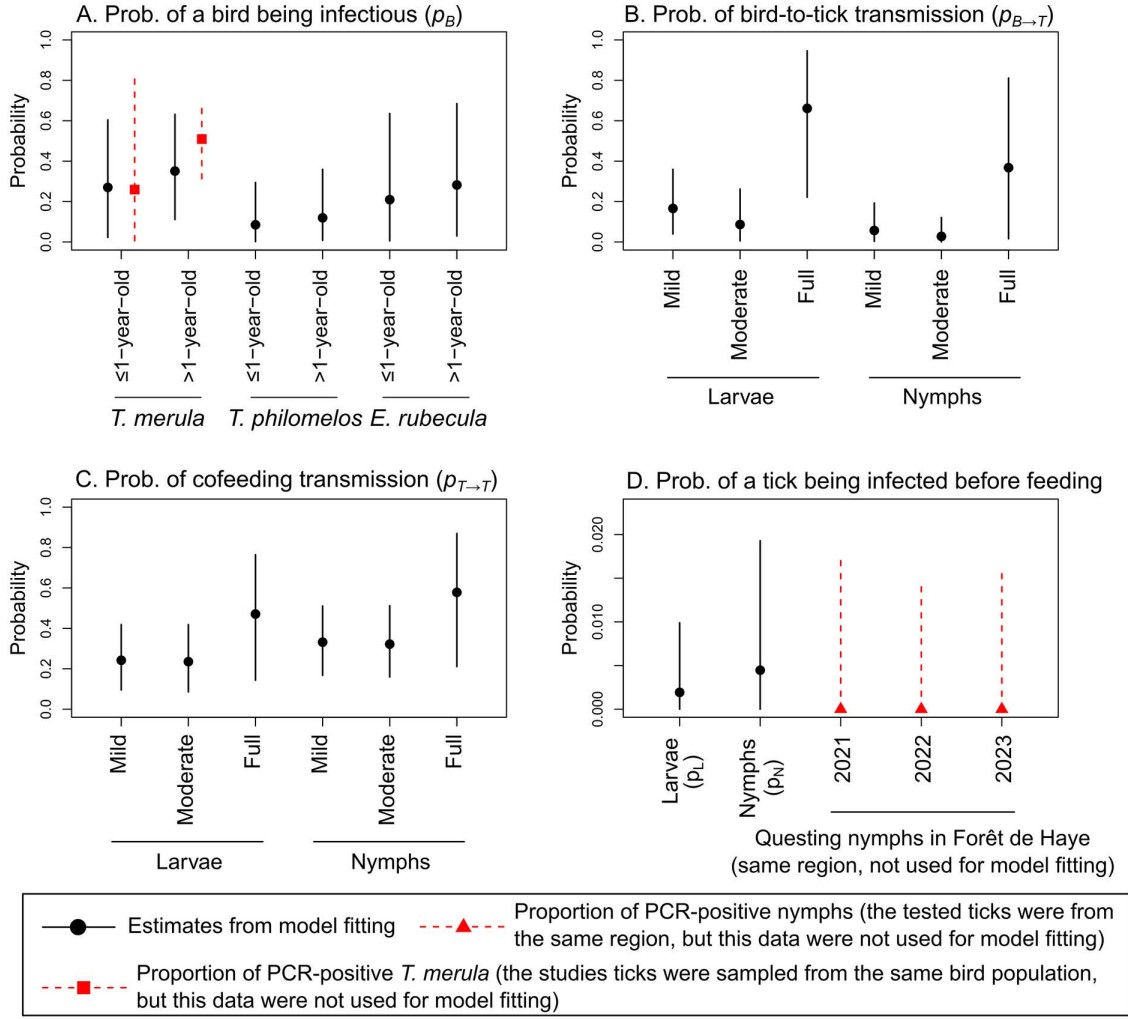

**Fig 6. Posterior parameter distribution from the full model fitted to the *A. phagocytophilum* dataset.** Black points and vertical lines represent the posterior median and 95%HDI estimates. Panel A shows the probability that a bird was infected and infectious by bird species and age group. Panel B shows the probability that a tick was infected from an infectious bird, by life stage and engorgement level. Panel C shows the probability that a tick was infected via cofeeding with other infected nymphs, by life stage and engorgement level. Panel D shows the probability that a tick was already infected before feeding, by life stage. Red triangles and dashed lines represent the proportion of *T. merula* birds that tested positive to *A. phagocytophilum*, from which the ticks in the *A. phagocytophilum* dataset were collected. This *T. merula* infection status data was not used for model fitting.

suggests that the transmission dynamics depend on both infection prevalence in ticks and birds and the aggregation of ticks on birds. Together, these factors determine the relative contribution of each transmission pathway and may help pathogen persistence under the varying conditions ticks encounter within the bird-tick system. These results also align with findings in the literature that demonstrated cofeeding transmission of these pathogens through experiments [2,32–36]. Our estimated probability of cofeeding transmission ($p_{T\to T}$) varied between the three pathogens studied, with its median ranging from 17% to 73%. Experimental studies have also reported varying levels of cofeeding transmission probabilities, for example, ranging 7% to 56% for *Borrelia burgdorferi* sensu stricto type [36], 0% to 55.3% for *Borrelia afzelii* [35], and 0–17% for *B. garinii* [33]. However, direct comparisons between these values and our estimates are not recommended, as they were derived from different study designs and different host, tick, and *Borrelia* species.

It should also be noted that, although cofeeding transmission is well-documented in the literature, it has primarily been studied independently of host-to-tick transmission and mostly in laboratory conditions. Similarly, most studies have investigated host-to-tick transmission without explicitly accounting for cofeeding transmission, reporting the prevalence only in ticks or hosts [37–40]. Importantly, the infection and immune status of hosts alone—if reliably available—is limited in providing meaningful insights into their role in pathogen transmission without integration with data on tick infestation and infection. In this context, our framework offers a robust foundation for simultaneously assessing these transmission routes and their relative contributions in the studied host and tick populations, leveraging readily available epidemiological data.

In this study, the model assuming transovarian transmission did not significantly improve a model fit compared to the model not assuming it, with its probability estimated extremely low. This suggests that larval infection in the data could be explained by other transmission routes, namely, bird-to-tick and cofeeding transmission. This result is consistent with the literature, which supports no or rare detection of the pathogens in *I. ricinus* larvae [41,42]. However, including this transmission route may be relevant for other pathogens, including the TBE virus [43].

Our simulation-based model assessment highlights that detailed information capturing variability in the likelihood of ticks feeding near infected nymphs is critical for improving the precision and reducing bias in parameter estimates for both bird-to-tick and cofeeding transmission pathways. Furthermore, we show that the availability of this information significantly influences how sample size affects the precision of these estimates. The importance of this specific information lies in its role within our framework where both transmission pathways are modelled jointly to predict the infection status of ticks presumed uninfected prior to feeding. That is, accounting for variability in cofeeding probabilities reduces uncertainty in cofeeding transmission likelihood, which, in turn, improves the identifiability of estimates for bird-to-tick transmission likelihood. This relationship is clearly demonstrated by comparing models: those incorporating heterogeneous probabilities of cofeeding with infected nymphs exhibited improved precision and reduced bias in parameter estimates compared to models assuming uniform probabilities, also achieving better precision at a given sample size.

These findings have important implications for field studies that sample hosts and their ticks to understand tick-borne disease transmission dynamics. In field studies, it is common practice to sample and test ticks that feed in aggregations on easily visible areas of the host's body. Thus, using the presence of at least one infected nymph as an indicator of whether cofeeding transmission is possible among ticks sampled from the same bird appears reasonable. During model fitting to the bird-tick pathogen data, we employed this approach prior to applying the actual cofeeding transmission probability ($p_{T \to T}$) to obtain the overall cofeeding transmission probability. However, it should be noted that, although not the majority, a non-negligible proportion of sampled birds (35.3%) had more ticks than were collected and tested, with engorged ticks preferentially sampled. This may have led to missed detections of infected nymphs, and depending on the association between tick engorgement and pathogen positivity and the clustering of infected ticks among others, our sampling design could have biased pathogen prevalence and cofeeding probability in ticks and downstream transmission probability estimates. Furthermore, given the importance of spatial distribution in cofeeding transmission [35], we acknowledge that the present approach did not fully account for spatial variability within tick aggregations. Furthermore, if infected nymphs had detached shortly before sampling, their potential contribution to cofeeding transmission might have been overlooked [35,44]. Additionally, the likely occurrence of small numbers of undetected feeding ticks was also not accounted for.

To address these limitations, future research applying our framework should consider incorporating additional ecological data and methodological approaches. First, at the time of capturing birds, spatial tick infestation data—including the exact feeding body site and, ideally, the precise location on that body site—should be collected. This information could help mapping the distribution of ticks on each body site, which, combined with tick infection data, would help refine the estimation of $p_{T \to T}$ and related parameters. Second, we recommend testing all identified ticks to provide a clearer understanding of bird–tick infestation and infection dynamics. In addition to efforts to gather more complete ecological and epidemiological data, data augmentation could be used to account for time-lagged contributions from non-sampled infected nymphs within a Bayesian inferential framework. The augmented data could include the number of ticks present within a

defined time window before sampling, along with their inferred infection status. The augmentation of these data could be informed by relevant ecological information, such as tick counts over multiple days on the same hosts through a series of capture and recaptures, as well as baseline infection prevalence in questing ticks.

Our model estimates from the bird-tick pathogen data fell within the ranges of values derived from the same data but not used during model fitting, as well as the data collected from the same forest area. First, for *A. phagocytophilum*, the estimated probability of birds being infectious was comparable to the proportion of birds whose blood samples tested positive, supporting the reliability of the model estimates. However, for *B. garinii* and *B. valaisiana*, their localised infection nature limited the utility of bird blood or tissue samples for directly comparing model estimates with actual host infection status, due to the relatively poor sensitivity of these sample types compared to using xenodiagnostic ticks [11]. Nonetheless, the support provided by the above comparison for *A. phagocytophilum* would also extend to the model estimates for *B. garinii* and *B. valaisiana*, as the analyses of these datasets were based on the same model and data structure. Furthermore, for all three pathogens, the consistency of the estimated probability of nymphs being infected before feeding with estimates from the same forest areas provides additional confidence in our model estimates. Most importantly, our simulation-based assessment demonstrated that our model can reliably recover true parameter values.

To validate the model estimates more directly for *Borrelia* infections, future studies applying the present framework could incorporate xenodiagnostics (e.g., using ticks free from the pathogen that experimentally fed on the host and then tested for the presence of the pathogen) to confirm host infectious status. A possible approach would involve performing xenodiagnostics for birds after their naturally feeding ticks are collected, while minimising *Borrelia* reactivation due to captivity-induced stress [12]. Zarka, Heylen [45] employed a novel study design combining xenodiagnostics with capture-recapture methods to minimise captivity-related stress, enabling a more reliable determination of infection status under natural conditions. In their study, 11% (n = 69) and 49% (n = 98) wild *Parus major* birds assessed in Northern Belgium transmitted *B. garinii* to at least one xenodiagnostics larva in breeding seasons of 2022 and 2023, respectively [45]. This range covers our estimated probability of a bird infectious, which was approximately 20% at the median. Although direct comparisons are limited by differences in bird species, sampling regions, and timing, the similar enzootic *Borrelia* situations in their study area and ours provide a valuable context for evaluating the reliability of our estimates.

In conclusion, we present a novel modelling framework that allows the assessment of tick-borne disease transmission dynamics at the tick-host interface, by explicitly describing transmission pathways and leveraging readily available epidemiological and ecological data in field studies that sample hosts and their ticks. Our simulation-based model assessment and comparison with actual field data support the reliability of parameter estimates derived from this framework given the underlying assumptions. In future studies, efforts to collect more detailed tick infestation data and combine xenodiagnostics data, while acknowledging substantial logistical and ecological challenges, may allow further refinement of this framework and its parameter estimates, to better capture tick-host-pathogen interactions.

## Supporting information

**S1 Text. Figs A to D.** Classification of tick life stage and engorgement level.
(DOCX)

**S1 Data. Supplementary data (Bird).** Supplementary data (Larva). Supplementary data (Nymph). Supporting Code.
(ZIP)

## Acknowledgments

We would like to express our gratitude to Professor Benoît Jaulhac at the French National Reference Center for Borrelia, University Hospital of Strasbourg, for providing valuable feedback on the manuscript. We would also like to express our gratitude to Stéphanie Etienne at ANSES for the analyses of pathogens in questing ticks.

## Author contributions

**Conceptualization:** Younjung Kim, Bruno Faivre, Thierry Boulinier, Laure Bournez, Raphaëlle Métras.

**Data curation:** Younjung Kim, Bruno Faivre, Célia Sineau, Clémence Galon, Sara Moutailler, Laure Bournez, Raphaëlle Métras.

**Formal analysis:** Younjung Kim, Clémence Galon, Sara Moutailler, Raphaëlle Métras.

**Funding acquisition:** Raphaëlle Métras.

**Investigation:** Younjung Kim, Bruno Faivre, Thierry Boulinier, Clémence Galon, Sara Moutailler, Laure Bournez, Raphaëlle Métras.

**Methodology:** Younjung Kim, Thierry Boulinier, Raphaëlle Métras.

**Project administration:** Raphaëlle Métras.

**Resources:** Clémence Galon, Sara Moutailler.

**Supervision:** Thierry Boulinier, Raphaëlle Métras.

**Validation:** Younjung Kim, Raphaëlle Métras.

**Visualization:** Younjung Kim.

**Writing – original draft:** Younjung Kim.

**Writing – review & editing:** Younjung Kim, Bruno Faivre, Thierry Boulinier, Célia Sineau, Clémence Galon, Sara Moutailler, Laure Bournez, Raphaëlle Métras.

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
