## [Decision Letter · Decision Letter 0]

25 Dec 2025

PCOMPBIOL-D-25-01943

A Bayesian Modelling Framework for Estimating Tick-Borne Pathogen Transmission Dynamics at the Host-Tick Interface

PLOS Computational Biology

Dear Dr. Kim,

Thank you for submitting your manuscript to PLOS Computational Biology. After careful consideration, we feel that it has merit but does not fully meet PLOS Computational Biology's publication criteria as it currently stands. Therefore, we invite you to submit a revised version of the manuscript that addresses the points raised during the review process.

We look forward to receiving your revised manuscript.

Kind regards,

Eric HY Lau, Ph.D.

Academic Editor

PLOS Computational Biology

Roger Kouyos

Section Editor

PLOS Computational Biology

**Additional Editor Comments:**

The Authors are expected to address all the criticisms by all Reviewers. In particular, please provide clearer justification of the specific host-vector components, host group, pathogens that were investigated (Reviewers #1, #2 & #3), provide more discussion of the quantitative insights, including how the estimated parameters translate into real-world implications, clarify “overdispersion”, provide a clear definition of “tested positive” (Reviewers #1 & #2), clarify sample collection period (Reviewers #1 & #3), assess the sensitivity of the model results to population size/demography (Reviewers #2 & #3) provide more description around sample size, provide more presentation of the results around the dataset and results, such as prevalences by species/age class, clarify the role of transovarial transmission, (Reviewer #1), clarified or discussed the minimum data requirement for the framework (Reviewer #2), provide a clear model formulation and calibration strategy and other methodological details (e.g. likelihood, free parameters, initial conditions, MCMC) (Reviewer #3).

**Journal Requirements:**

4) Please amend your detailed Financial Disclosure statement. This is published with the article. It must therefore be completed in full sentences and contain the exact wording you wish to be published.

**Reviewers' comments:**

Reviewer's Responses to Questions

**Comments to the Authors:**

Reviewer #1: A Bayesian Modelling Framework for Estimating Tick-Borne Pathogen Transmission Dynamics at the Host-Tick Interface

Authors

Younjung Kim, Bruno Faivre, Thierry Boulinier, Célia Sineau, Clémence Galon, Sara Moutailler, Laure Bournez, Raphaëlle Métras

General comments:

The study develops an original framework that successfully isolates different transmission pathways using ecological and epidemiological field data. By combining simulations with applications to bird-tick datasets for three pathogens, the authors demonstrate that their approach reliably retrieves key parameters. The work further shows that incorporating fine-scale tick interaction data greatly strengthens inference. Overall, the framework represents a valuable and innovative contribution with clear practical relevance for designing future field studies.

Introduction

The introduction would benefit from clearer justification of the specific host-vector components investigated. The current text does not explain why the analysis focuses on bird-to-tick or tick-to-bird transmission, nor why birds were chosen as the focal host group in the first place. Similarly, the rationale for selecting the three pathogens, Ixodes ricinus, and the three bird species is not sufficiently developed. Providing broader context on Lyme disease ecology, particularly the roles of vertebrate hosts, the contribution of bird hosts, and the diversity of tick species and associated pathogens, would help readers understand the relevance of these choices. In addition, the discussion of sample size is ambiguous: the introduction does not specify whether sample size refers to numbers of hosts, ticks, or both, and clarifying this point early on would improve the reader’s understanding of the study’s objectives.

Materials and Methods

The Materials and Methods section is generally clear, well structured, and technically well argued. However, some important aspects remain implicit or insufficiently explained. For instance, it is unclear whether the model accounts for transovarial transmission, and (again) the examination of sample size effects is not explicitly defined. In addition, the dataset description lacks basic information such as the period of collection, which would help contextualize the analyses. Overall, the methodological framework would be strengthened by anchoring it more firmly in biological assumptions, for example by explicitly stating hypotheses related to sample size or overdispersion. This would improve interpretability and help readers connect the modelling choices to underlying biological processes.

Results

The Results section would benefit from presenting key empirical patterns from the dataset. In particular, reporting observed pathogen prevalence by bird species and age class, as well as prevalence in questing ticks and across tick developmental stages, would provide valuable context and help readers assess how the model outputs relate to the underlying data.

Discussion

The Discussion would benefit from addressing how the choice of bird species, the geographic origin of the field data, and the (unreported) sampling period may influence the results and their generality. More importantly, the section lacks a broader interpretation of the quantitative insights that the proposed framework can provide, particularly regarding the contribution of birds to the circulation of the studied pathogens. Placing the findings in this ecological and epidemiological context would substantially strengthen the discussion.

Specific comments:

Abstract

Line 26: The phrase “estimated key transmission parameters and associated factors” is too vague. Please specify which parameters and which associated factors are being estimated.

Line 29: The “impact of sample size” needs clarification. Does sample size refer to numbers of hosts, ticks, feeding events, or another unit?

Lines 30-31: The statement “field data collected at the bird-tick interface” lacks essential contextual details. Please indicate where and when these data were collected.

Line 35: The sentence about improved performance when more information on variability in cofeeding probabilities was available should be expanded. What are the practical implications of this finding for field data collection and future research? As this is a key result, readers need to understand its consequences.

Lines 39-40: The claim that “Cofeeding transmission was found as a significant factor for all three pathogens” is too imprecise. Please indicate for which outcome or parameter cofeeding transmission was significant.

Introduction

Line 63: The focus on “bird-to-tick or tick-to-bird” transmission is not sufficiently justified. Other hosts play important roles in the system, and the introduction should explain why the study focuses specifically on birds.

Line 110: The phrase “assess the effect of sample size” needs clarification. Specify whether this refers to the number of hosts, ticks, feeding events, or another sampling unit.

Lines 112-116: The introduction should better contextualize the selected hosts, tick species, and pathogens, and justify why these particular organisms were chosen. In particular, the relevance of including Borrelia valaisiana, which typically shows low prevalence, should be addressed.

Materials and Methods

Lines 131-133: The sentence beginning “The underlying framework… small mammals” presents a forward-looking perspective. This type of content belongs in the Discussion section.

Lines 245-246: The statement “Nymphs are assumed to have a higher probability than larvae” requires clarification regarding transovarial transmission. How is transovarial transmission from infected females to larvae taken into account in the model? Are larvae initialized with zero infection, or with a non-zero baseline prevalence? If so, what value is used, and does it vary by pathogen?

Line 253: The term “sample size” again needs to be specified.

Line 255: The phrase “to assess the impact of overdispersion of the data (ticks are clustered by birds)” is too concise. Explain briefly how overdispersion is expected to influence parameter precision and why this is important for the system.

Line 265: The text refers to “mean… empirical data” but does not provide the corresponding results. Please report these empirical values.

Line 268: Explain why only Borrelia garinii was used at this step.

Line 269: The expression “using randomly selected parameter values” is unclear. Specify how these parameters were chosen, from which ranges, and for what purpose.

Paragraph 2: Impact of sample size: As previously noted, define which sample size is being examined. Also, articulate any biological or statistical hypotheses regarding how sample size is expected to affect the estimates.

Paragraph 3: Impact of tick overdispersion: Similarly, clarify the hypotheses being tested. What effects do you expect overdispersion (tick aggregation on hosts) to have on estimation accuracy?

Line 297: “for tick distributions” should be clarified: are these distributions on birds, on questing ticks, or both?

Line 304: Indicate the period of collection (months/season and year). Tick phenology and pathogen prevalence vary strongly with time.

Line 310: Erithacus rubecula is no longer part of Turdidae. Also correct the spelling: remove the final “r” in rubecular.

Line 313: The statement “on the head and legs, where ticks are more found” should be supported with a reference. Head infestation is common, but high tick loads on legs are not commonly reported.

Line 317: Justify the decision to collect a maximum of only 10 ticks per bird. How might this choice affect prevalence estimates or parameter inference?

Line 334: Again, specify the collection period. The timing strongly influences observed prevalence and should be reported.

Results

Figure 2.E: The title contains a typo: replace “larave” with “larvae”.

Line 398: Replace “studies” with “studied.”

Line 436: The statement “with no significant difference observed between models that did or did not assume prior infection at the larval stage” requires more explanation. Please clarify: (i) how transovarial transmission was incorporated (or not) into the model, (ii) whether larvae were initialized with zero or non-zero infection levels, (iii) and what the literature reports regarding expected larval infection prevalence for each of the three pathogens studied.

Figure 5: The same symbol (red triangle) appears to represent different quantities in panels A and D, creating confusion. Please modify the symbols and legend to avoid ambiguity.

Figure 5.A: Explain why observational data are shown only for Turdus merula and not for the two other bird species.

Figure 5 – legend: The phrase “proportion of T. merula birds that tested positive to A. phagocytophilum” is ambiguous. Does “tested positive” mean that at least one tick collected from a bird was infected? If so, please state this clearly; if not, specify the criterion used.

Discussion

Line 498: The term “critical factor” is vague. Rather than using qualitative language, explain why cofeeding is important for transmission in this system. What does this result imply? For instance, does the model suggest that cofeeding contributes a substantial, and highly variable, proportion of overall transmission? Clarifying this would help readers understand the biological significance of the finding.

Line 518: The statement that the framework “could provide more realistic estimates” should be nuanced. The empirical datasets used here are relatively small and restricted to a few bird species and only one site, which limits the generality of the conclusions. Acknowledge these constraints when discussing realism.

Lines 403-406: The explanation for this result is missing. Please expand on the biological or methodological mechanisms that could account for this pattern.

Lines 425-428: The interpretation of these results should be placed in the context of the field prevalence of the three pathogens. Without this perspective, it is difficult to assess how representative or meaningful the modelled effects are.

References

Several references contain typographical errors or missing letters (e.g., urdus instead of Turdus, orrelia instead of Borrelia in the reference number 13). A thorough revision of the reference list is needed to correct these types of errors.

Reviewer #2: This manuscript presents a well-structured and scientifically compelling modelling framework for investigating the transmission dynamics of tick-borne pathogens at the host–tick interface. By integrating multiple infection pathways within a Bayesian inference approach informed by field data, the study provides a robust quantitative basis for identifying key drivers of pathogen transmission. The manuscript is clearly written, methodologically rigorous, and conceptually engaging. In a few sections, the narrative could be streamlined to enhance the practical applicability of the framework, and the inclusion of concrete examples could help readers better understand how the estimated parameters translate into real-world implications. Overall, the manuscript fits well within the scope of the journal and constitutes a valuable contribution to the field.

Minor comments below

Line 34. I am not fully convinced by the use of the word “true”; “realistic” may be more appropriate in this context?

Lines 52–53 (and similarly line 68). It is very interesting that the framework is presented as applicable to routinely collected field data. However, I wonder whether host-based tick sampling can truly be considered routine. In many settings, routine surveillance more commonly involves collecting ticks from the environment, whereas host-derived ticks may be less frequently available and more challenging to obtain, particularly for certain wildlife species. I would suggest clarifying or slightly expanding this point to acknowledge these practical differences while still emphasising the value of the proposed approach.

Line 54. The manuscript does not explain why authors chose these three systems instead of others. Providing a brief biological rationale would strengthen the argument and improve replicability.

Lines 76–77. I am not entirely sure that obtaining all the required information—especially age-structured sampling, for instance—is straightforward in wildlife studies or routine surveillance. A similar consideration applies to lines 127–128. Could author expand including this limitation or explaining why this is not a limitation for replicability of the study?

Line 103. It is unclear whether infection risk refers to the risk for birds, ticks, or both. Clarifying this would improve readability.

Line 130. I would recommend expanding this section by briefly explaining the rationale for choosing this particular system. What features make it a suitable case study, and which characteristics support generalisation to other host–tick systems? Conversely, are there systems where this framework would not be applicable? For instance, line 133 mentions the possibility of extending it to small mammals, why specifically small mammals, and not mammals more broadly?

Line 162. It may be helpful to clearly state which bird-specific characteristics need to be considered in the modelling framework, to guide readers who may be less familiar with the system or this kind of modelling.

Line 245. It would be useful to clarify what is meant by higher in this context. Does it refer to the constraint that nymphal pTinf must just somehow exceed that of larvae, or does it imply something else?

Line 263. Is there any information available on the sensitivity of the model to population size? A brief comment on this would strengthen the interpretation of the results.

Line 286. It is not entirely clear which model is being referred to here when you write “a model”. Providing a more explicit reference would improve clarity.

Lines 288–290. Could you clarify why the system showing a significant association between the presence of at least one infected nymph and the estimated number of ticks was selected for this analysis?

Line 317. Could the authors expand on the rationale behind this choice, and behind selecting the value of 10?

Line 330. Do the two methods yield completely identical results? If not, how comparable are they? Why did the authors chose this method? A brief comparison would strengthen this section.

Line 332. Is there a way to (even roughly) standardise the tick classification used here? For applying this framework to other datasets, it would be helpful to know whether the same categories can be used and how they might be replicated.

Line 334. For the sake of reproducibility, it may be beneficial to add a brief description of how these data were collected.

Line 368. The captions of Figures S1 and S2 could be improved. At present, the figures are not very clear, and the captions are not sufficiently self-explanatory. Providing a few additional details would help readers interpret the results more easily.

Line 373. I am not entirely sure which overdispersion scenario the authors are referring to here. Earlier in the manuscript, two overdispersion scenarios were mentioned, could this be clarified?

Figure 1. The figure is not fully clear to me in its current form. In particular, I do not see where sample size is analysed within the figure, and it is unclear why the Poisson scenario appears only in panel C. Shouldn’t the Poisson assumption influence the estimation of other parameters as well?

Line 413. Do the authors have any suggestions regarding a possible biological explanation for this result? Even a brief hypothesis would help contextualise the finding.

Figures 3 and 4. It would be helpful to guide the reader more explicitly in interpreting these figures. The captions describe what is shown, but provide little indication of what the patterns mean or how they should be interpreted. Adding interpretative guidance would greatly aid the reader’s understanding.

The discussion is overall complete and very well written; I only have a few minor points to raise:

Line 470. It would be helpful if the authors could more clearly describe the datasets used. For a biologist interested in applying a similar framework, having a clearer idea of the minimum data requirements would be extremely valuable. Given the strength of the work, I would also suggest briefly outlining in which situations this framework can be applied, and any limitations, to encourage cross-disciplinary uptake of the framework.

Line 478. As above, further clarification of the data requirements and potential constraints would strengthen the practical relevance of the work.

Line 540. The choice of using only 10 ticks, and the implications of this choice,remains not entirely clear to me. A short explanation would be useful.

Line 597. It may be worth acknowledging somewhere that studies of this type require substantial time, effort, and resources.

Reviewer #3: The authors present an interesting modelling framework to investigate the transmission dynamics of tick-borne pathogens and apply it to field data to estimate several epidemiologically relevant parameters. Overall, the study is of interest and the manuscript is clearly written, although several points require further clarification or detail.

I suggest including a schematic representation of the model structure to help readers grasp its components, as well as providing the full set of governing equations. It appears that demographic processes (such as tick and host fecundity and survival) are not included; could you clarify the rationale for omitting them? I also find it unclear why some baseline infection probabilities (e.g., α_pB, pTinf) are assumed rather than being generated by explicit infection processes. This may relate to lines 175–179, which would benefit from clearer explanation.

The model includes only one host category, namely birds. As far as I know, birds are generally not the most competent reservoirs for these pathogens compared with rodents or other mammals. Could you elaborate on the justification and implications of this choice?

Equations 9 and 10 also require further clarification. FOI1 is described as the probability that birds become infected via cofeeding and/or transmission; what then is FOI2?

Lines 345–361 need additional methodological detail. Please provide the explicit likelihood formula and the complete list of free parameters that are estimated. What initial conditions were used? What was the simulated time span? How many MCMC iterations were run, and how was convergence assessed? I also suggest moving Figure S3 into the main text. Its caption should likely read “Model fit”.

Minor points

Please provide some additional details on Borrelia and Anaplasma in the introduction upon their first mention, such as their public health burden (possibly with some numbers of cases), range of competent hosts and vectors.

When were birds sampled?

Please explain what HDI stands for upon its first appearance.

I’d have expected some plots showing estimated probabilities as functions. Please clarify what you mean by characteristics for vectors β (eq.1, 3, 8) and list the full coefficients lists. Are they free parameters to be estimate via MCMC? If so, this connects directly to the main methodological questions above.

Please note that in the provided R code there is a “setwd” on line 185 which I believe ought to be removed.

**Have the authors made all data and (if applicable) computational code underlying the findings in their manuscript fully available?**

The PLOS Data policy requires authors to make all data and code underlying the findings described in their manuscript fully available without restriction, with rare exception (please refer to the Data Availability Statement in the manuscript PDF file). The data and code should be provided as part of the manuscript or its supporting information, or deposited to a public repository. For example, in addition to summary statistics, the data points behind means, medians and variance measures should be available. If there are restrictions on publicly sharing data or code —e.g. participant privacy or use of data from a third party—those must be specified.requires authors to make all data and code underlying the findings described in their manuscript fully available without restriction, with rare exception (please refer to the Data Availability Statement in the manuscript PDF file). The data and code should be provided as part of the manuscript or its supporting information, or deposited to a public repository. For example, in addition to summary statistics, the data points behind means, medians and variance measures should be available. If there are restrictions on publicly sharing data or code —e.g. participant privacy or use of data from a third party—those must be specified.requires authors to make all data and code underlying the findings described in their manuscript fully available without restriction, with rare exception (please refer to the Data Availability Statement in the manuscript PDF file). The data and code should be provided as part of the manuscript or its supporting information, or deposited to a public repository. For example, in addition to summary statistics, the data points behind means, medians and variance measures should be available. If there are restrictions on publicly sharing data or code —e.g. participant privacy or use of data from a third party—those must be specified.requires authors to make all data and code underlying the findings described in their manuscript fully available without restriction, with rare exception (please refer to the Data Availability Statement in the manuscript PDF file). The data and code should be provided as part of the manuscript or its supporting information, or deposited to a public repository. For example, in addition to summary statistics, the data points behind means, medians and variance measures should be available. If there are restrictions on publicly sharing data or code —e.g. participant privacy or use of data from a third party—those must be specified.

Reviewer #1: Yes

Reviewer #2: Yes

Reviewer #3: Yes

PLOS authors have the option to publish the peer review history of their article (what does this mean?). If published, this will include your full peer review and any attached files.). If published, this will include your full peer review and any attached files.). If published, this will include your full peer review and any attached files.). If published, this will include your full peer review and any attached files.

...

Reviewer #1: No

Reviewer #2: No

Reviewer #3: No

**Figure resubmission:**

---

## [Decision Letter · Decision Letter 1]

20 Mar 2026

Dear Dr Kim,

We are pleased to inform you that your manuscript 'A Bayesian Modelling Framework for Estimating Tick-Borne Pathogen Transmission Dynamics at the Host-Tick Interface' has been provisionally accepted for publication in PLOS Computational Biology.

Best regards,

Eric HY Lau, Ph.D.

Academic Editor

PLOS Computational Biology

Roger Kouyos

Section Editor

PLOS Computational Biology

Thanks for addressing all the reviewers' comments. Congratulations on the excellent work!

Reviewer's Responses to Questions

**Comments to the Authors:**

Reviewer #1: All my comments have been addressed following the authors’ review of the article; therefore, I have no further recommendations to provide.

Reviewer #3: Thank you for addressing all my previous comments and concerns

**Have the authors made all data and (if applicable) computational code underlying the findings in their manuscript fully available?**

The PLOS Data policy requires authors to make all data and code underlying the findings described in their manuscript fully available without restriction, with rare exception (please refer to the Data Availability Statement in the manuscript PDF file). The data and code should be provided as part of the manuscript or its supporting information, or deposited to a public repository. For example, in addition to summary statistics, the data points behind means, medians and variance measures should be available. If there are restrictions on publicly sharing data or code —e.g. participant privacy or use of data from a third party—those must be specified.requires authors to make all data and code underlying the findings described in their manuscript fully available without restriction, with rare exception (please refer to the Data Availability Statement in the manuscript PDF file). The data and code should be provided as part of the manuscript or its supporting information, or deposited to a public repository. For example, in addition to summary statistics, the data points behind means, medians and variance measures should be available. If there are restrictions on publicly sharing data or code —e.g. participant privacy or use of data from a third party—those must be specified.requires authors to make all data and code underlying the findings described in their manuscript fully available without restriction, with rare exception (please refer to the Data Availability Statement in the manuscript PDF file). The data and code should be provided as part of the manuscript or its supporting information, or deposited to a public repository. For example, in addition to summary statistics, the data points behind means, medians and variance measures should be available. If there are restrictions on publicly sharing data or code —e.g. participant privacy or use of data from a third party—those must be specified.requires authors to make all data and code underlying the findings described in their manuscript fully available without restriction, with rare exception (please refer to the Data Availability Statement in the manuscript PDF file). The data and code should be provided as part of the manuscript or its supporting information, or deposited to a public repository. For example, in addition to summary statistics, the data points behind means, medians and variance measures should be available. If there are restrictions on publicly sharing data or code —e.g. participant privacy or use of data from a third party—those must be specified.

Reviewer #1: Yes

Reviewer #3: None

PLOS authors have the option to publish the peer review history of their article (what does this mean?). If published, this will include your full peer review and any attached files.). If published, this will include your full peer review and any attached files.). If published, this will include your full peer review and any attached files.). If published, this will include your full peer review and any attached files.

...

Reviewer #1: No

Reviewer #3: No

---

## [Editor Report · Acceptance letter]

PCOMPBIOL-D-25-01943R1

A Bayesian Modelling Framework for Estimating Tick-Borne Pathogen Transmission Dynamics at the Host-Tick Interface

Dear Dr Kim,

I am pleased to inform you that your manuscript has been formally accepted for publication in PLOS Computational Biology. Your manuscript is now with our production department and you will be notified of the publication date in due course.

With kind regards,

Judit Kozma
